# Digital adherence technology for tuberculosis treatment supervision: A stepped-wedge cluster-randomized trial in Uganda

Adithya Cattamanchi[1,2☯]*, Rebecca Crowder[1☯], Alex Kityamuwesi[2☯],
Noah Kiwanuka[3], Maureen Lamunu[2], Catherine Namale[2], Lynn Kunihira Tinka[2],
Agnes Sanyu Nakate[2], Joseph Ggita[2], Patricia Turimumahoro[2], Diana Babirye[2],
Denis Oyuku[2], Christopher Berger[1], Austin Tucker[4], Devika Patel[5],
Amanda Sammann[5], Stavia Turyahabwe[6], David Dowdy[2,4], Achilles Katamba[2,7]

1 Center for Tuberculosis and Division of Pulmonary and Critical Care Medicine, San Francisco General
Hospital, University of California San Francisco, San Francisco, California, United States of America,
2 Uganda Tuberculosis Implementation Research Consortium, Walimu, Kampala, Uganda, 3 Department of
Epidemiology and Biostatistics, School of Public Health, Makerere University College of Health Sciences,
Kampala, Uganda, 4 Department of Epidemiology, Johns Hopkins Bloomberg School of Public Health,
Baltimore, Maryland, United States of America, 5 The Better Lab, Department of Surgery, San Francisco
General Hospital, University of California San Francisco, San Francisco, California, United States of America,
6 National Tuberculosis and Leprosy Program, Uganda Ministry of Health, Kampala, Uganda, 7 Clinical
Epidemiology and Biostatistics Unit, Department of Medicine, Makerere University College of Health
Sciences, Kampala, Uganda

☯ These authors contributed equally to this work.
* adithya.cattamanchi@ucsf.edu

pmed.1003628

Medicine Editorial Board, UNITED STATES

**Data Availability Statement:** All relevant data are
within the manuscript and its Supporting
Information files.

## Abstract

### Background

Adherence to and completion of tuberculosis (TB) treatment remain problematic in many
high-burden countries. 99DOTS is a low-cost digital adherence technology that could
increase TB treatment completion.

### Methods and findings

We conducted a pragmatic stepped-wedge cluster-randomized trial including all adults
treated for drug-susceptible pulmonary TB at 18 health facilities across Uganda over 8
months (1 December 2018–31 July 2019). Facilities were randomized to switch from routine
(control period) to 99DOTS-based (intervention period) TB treatment supervision in conse-
cutive months. Patients were allocated to the control or intervention period based on which
facility they attended and their treatment start date. Health facility staff and patients were not
blinded to the intervention. The primary outcome was TB treatment completion. Due to the
pragmatic nature of the trial, the primary analysis was done according to intention-to-treat
(ITT) and per protocol (PP) principles. This trial is registered with the Pan African Clinical Tri-
als Registry (PACTR201808609844917). Of 1,913 eligible patients at the 18 health facilities
(1,022 and 891 during the control and intervention periods, respectively), 38.0% were
women, mean (SD) age was 39.4 (14.4) years, 46.8% were HIV-infected, and most (91.4%)
had newly diagnosed TB. In total, 463 (52.0%) patients were enrolled on 99DOTS during

**Funding:** This Project is supported by the Stop TB Partnership's TB REACH initiative, grant number STBP/TBREACH/GSA/W6-37 (AC, AKa), which is funded by the Government of Canada, the Bill & Melinda Gates Foundation, and the United States Agency for International Development. The funder had no role in study design, data collection and analysis, decision to publish, or preparation of the manuscript.

**Competing interests:** I have read the journal's policy and the authors of this manuscript have the following competing interests: AS is owner and consultant for the human-centered design consultancy, The Empathy Studio, LLC. DP is a human-centered design consultant for The Empathy Studio, LLC. The other authors have declared that no competing interests exist.

**Abbreviations:** aOR, adjusted odds ratio; DAT, digital adherence technology; DOT, directly observed therapy; ITT, intention-to-treat; NTLP, National TB and Leprosy Program; PP, per protocol; TB, tuberculosis; WHO, World Health Organization.

the intervention period. In the ITT analysis, the odds of treatment success were similar in the intervention and control periods (adjusted odds ratio [aOR] 1.04, 95% CI 0.68–1.58, $p = 0.87$). The odds of treatment success did not increase in the intervention period for either men (aOR 1.24, 95% CI 0.73–2.10) or women (aOR 0.67, 95% CI 0.35–1.29), or for either patients with HIV infection (aOR 1.51, 95% CI 0.81–2.85) or without HIV infection (aOR 0.78, 95% CI 0.46–1.32). In the PP analysis, the 99DOTS-based intervention increased the odds of treatment success (aOR 2.89, 95% CI 1.57–5.33, $p = 0.001$). The odds of completing the intensive phase of treatment and the odds of not being lost to follow-up were similarly improved in PP but not ITT analyses. Study limitations include the likelihood of selection bias in the PP analysis, inability to verify medication dosing in either arm, and incomplete implementation of some components of the intervention.

## Conclusions

99DOTS-based treatment supervision did not improve treatment outcomes in the overall study population. However, similar treatment outcomes were achieved during the control and intervention periods, and those patients enrolled on 99DOTS achieved high treatment completion. 99DOTS-based treatment supervision could be a viable alternative to directly observed therapy for a substantial proportion of patients with TB.

## Trial registration

Pan-African Clinical Trials Registry (PACTR201808609844917).

---

Author summary

### Why was this study done?

- A PubMed search for publications prior to 1 January 2019 with the search terms "adherence technology AND tuberculosis" revealed 4 randomized trials and 14 observational studies of digital adherence technologies (DATs) being used to support tuberculosis (TB) treatment, most of which focused primarily on adherence rather than treatment outcome.

- A systematic review found limited evidence to support the effectiveness of DATs for improving TB treatment outcomes.

### What did the researchers do and find?

- We adapted 99DOTS, a low-cost DAT already widely used in India, with input from local stakeholders, and conducted a pragmatic randomized trial of the resulting 99DOTS-based intervention at 18 health facilities in Uganda.

- Only about half of patients were initiated on 99DOTS-based treatment supervision during the intervention period.

- The 99DOTS-based intervention did not increase treatment completion in the full study population.

- Treatment completion was high (>85%) among the nonrandom sample of patients initiated on 99DOTS-based treatment supervision during the intervention period.

### What do these findings mean?

- 99DOTS should not be used as a universal replacement for directly observed therapy for TB treatment supervision, with the aim of increasing population-level treatment completion.

- 99DOTS-based treatment supervision could enable a substantial proportion of patients with TB to complete treatment without the inconvenience and additional costs of directly observed therapy.

- Further research is needed to assess whether overall treatment outcomes can be improved by increasing uptake of 99DOTS or other low-cost DATs, and to identify additional measures needed to support all patients to complete treatment.

## Introduction

Ensuring that patients complete treatment for tuberculosis (TB), the leading infectious cause of death worldwide, remains a key challenge to achieving cure in many high-burden countries [1]. Since 1993, a health worker observing the patient when he or she swallows each dose of anti-TB medication (directly observed therapy [DOT]) has been a central aspect of the World Health Organization (WHO)–recommended strategy for TB treatment supervision [2]. However, DOT is time-consuming and costly for patients and health workers, and multiple trials have failed to demonstrate improvement in treatment outcomes [3]. Novel, patient-centered approaches are needed to monitor and promote TB treatment adherence and completion.

Recently, there has been increasing interest in digital adherence technologies (DATs) as an alternative to DOT. DATs enable patients to take TB medicines at home, restoring patient autonomy and dignity, while still enabling health workers to monitor and support adherence [4]. A common DAT platform generates real-time adherence data by placing medications within an electronic pill box. In the context of TB treatment, a cluster-randomized trial in China demonstrated improved adherence with medication reminders delivered via an electronic pill box. However, the trial was not powered to evaluate a difference in TB treatment outcomes. 99DOTS (Everwell Health Solutions, India) is an alternative low-cost DAT that involves patients calling toll-free phone numbers that are ordered in an unpredictable pattern and hidden underneath pills in blister packs [5,6]. The phone numbers are revealed only when patients remove scheduled medication doses from the blister pack, enabling patients to make toll-free calls to self-report medication dosing. Health facility staff can access adherence data for individual patients in real-time through a web dashboard and mobile phone application. Although 99DOTS has been widely scaled up in India, it has not yet been rigorously evaluated in any country. High-quality evidence of impact on treatment outcomes is needed to support WHO policy recommendations [7] and further uptake in other high-TB-burden countries.

The DOT to DAT trial aimed to determine whether a 99DOTS-based treatment supervision strategy improved TB treatment completion in comparison to routine TB treatment supervision. We secondarily assessed the reach of the intervention strategy (proportion initiated on 99DOTS) and short-term treatment outcomes (persistence on treatment through the intensive phase; loss to follow-up). The cluster-randomized design allowed us to study the 99DOTS-based intervention as it would be used in routine care. The stepped-wedge design increased acceptability among the participating health facilities and feasibility of training sites on the intervention. 99DOTS implementation and health economic outcomes are described in the trial protocol [8]; implementation and health economic outcomes will be reported later. Previous publications have described the trial protocol [8], baseline TB treatment outcomes at the trial sites [9], and the human-centered design process used to adapt the 99DOTS platform to better meet the needs of patients and health workers [10,11].

## Methods

### Study design and participants

We conducted a pragmatic stepped-wedge cluster-randomized trial of a 99DOTS-based TB treatment supervision strategy at 18 health facilities in Uganda with National TB and Leprosy Program (NTLP)–affiliated TB treatment units. The 18 health facilities included 5 regional referral hospitals, 10 general hospitals, and 3 district health centers. Health facilities were eligible for the trial if they treated >10 TB patients/month in 2017, were located within 225 of Kampala but not within Kampala District, and had a TB treatment success rate < 80% in 2017 (S1 Fig).

At each of the 18 health facilities, we included data from all adults (age ≥ 18 years) who initiated treatment for drug-susceptible pulmonary TB between 1 December 2018 and 31 July 2019 and followed patients passively until a TB treatment outcome was assigned. Patients who were transferred to another treatment unit were excluded from the study population, as they would not have had an opportunity to receive the intervention throughout treatment (S1 Fig).

The trial followed a repeated cross-sectional design, with each 1-month period capturing different patients initiated on TB treatment (S2 Fig). All health facilities started with routine TB treatment supervision (control period). Subsequently, 3 health facilities per month were randomly switched to 99DOTS-based TB treatment supervision (intervention period) over a 6-month period. Patients were allocated to the intervention or control period based on the facility they attended and the month in which they started treatment; patients who started treatment when their health facility was in the control period were therefore ineligible to receive 99DOTS-based treatment supervision for the duration of their treatment. The first month of switching to the intervention was considered to be a transition (i.e., buffer) period, during which health facility staff were trained on how to use 99DOTS. Patients who started TB treatment during the buffer period for each health facility were excluded from the primary analysis.

The trial was approved by institutional review boards at Makerere University School of Public Health and the University of California San Francisco, and by the Uganda National Council for Science and Technology. A waiver of informed consent was granted to access patient demographic and clinical information recorded in TB treatment registers. The trial protocol has been published previously [8], and copies of the trial protocol, statistical analysis plan, and CONSORT checklist are available as S1 Trial Protocol, S1 Statistical Analysis Plan, and S1 CONSORT Checklist, respectively.

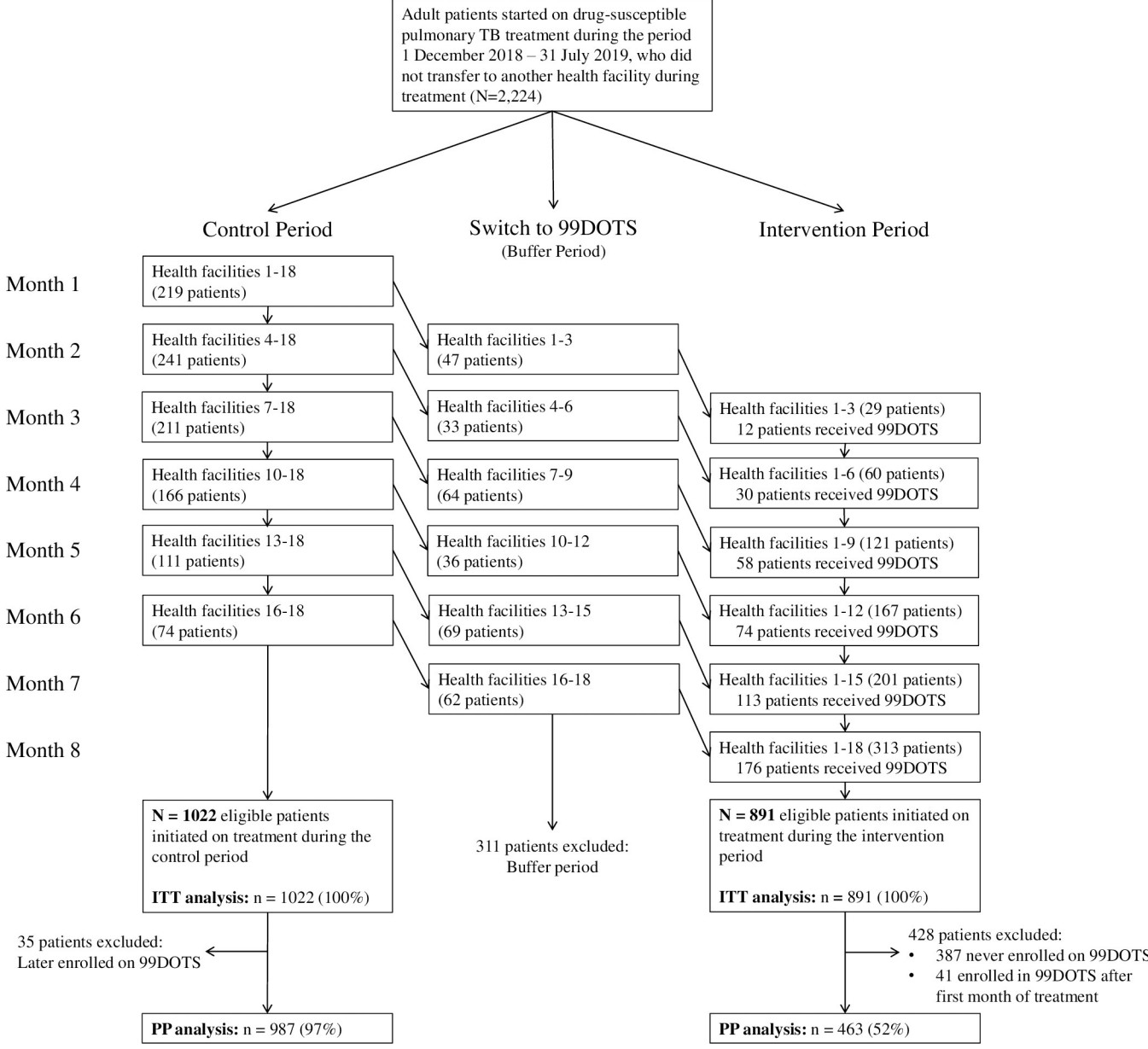

**Fig 1. Trial profile.** ITT, intention-to-treat; PP, per protocol; TB, tuberculosis.

## Randomization and masking

Before the trial began, the 18 health facilities were first assigned randomly to 1 of 6 blocks of 3 health facilities, and then the order in which each block switched from routine care to the intervention was also assigned randomly using a simple, unrestricted 2-stage process (S2 Fig). The randomization process was carried out at a ceremony attended by stakeholders from all participating health facilities in which facility representatives participated in the randomization by pulling labeled balls out of a bag, determining their allocation [8]. Blinding of the intervention time period to health facility staff and patients was not feasible as they were the targets of the intervention. Investigators and study staff, with the exception of the biostatistician (NK) and data manager (RC), were blinded to treatment outcomes.

## Procedures

Staff at each health facility took photos of health facility TB treatment registers and uploaded the photos monthly to a secure server. Research staff then abstracted demographic, clinical, and outcome data for patients initiated on TB treatment from the photos and entered the data into a Research Electronic Data Capture (REDCap) database [8]. Missing data for key variables in treatment registers were obtained via phone call with health facility staff.

During the control period, health facilities continued providing TB treatment supervision in the same manner as before the trial. The community-based DOT model used at the 18 health facilities has been described previously, including the substantial variation in how TB treatment supervision is implemented across the facilities [9]. All facilities request patients to take their TB medicines under direct observation by a treatment supporter. The treatment supporter is most commonly a family member who receives no training or compensation. All health facilities report assessing adherence (most commonly by patient self-report) at drug refill visits scheduled every 2 weeks during the intensive phase and every 4 weeks during the continuation phase. Patients who miss refill visits are not followed up at the majority of health facilities.

During the buffer period for each health facility, research staff conducted a 3-day training on the 99DOTS-based intervention at the facility [8]. Following the training, health facility staff were requested to offer 99DOTS-based TB treatment supervision to all adults initiating treatment for pulmonary TB at their facility. Patients who had previously initiated treatment continued to receive routine care as described above.

During the intervention period, the decision to offer and accept 99DOTS-based supervision was left to treating providers and patients, respectively. The standard 99DOTS platform was adapted using human-centered design methods in collaboration with local stakeholders and end-users prior to the start of the trial, with further adaptation of the envelope design during the first 2 months of the trial (S3 Fig) [10,11]. If enrolled on 99DOTS, patients were given TB medication blister packs placed inside the adapted 99DOTS envelope, received daily automated SMS dosing reminders, and were asked to confirm dosing by making daily toll-free phone calls to the 99DOTS system. Patients heard a rotating series of educational and motivational messages when they called in to report dosing. As in the control period, drug refill visits were scheduled every 2 weeks during the intensive phase and every 4 weeks during the continuation phase.

## Outcomes

Treatment outcomes were recorded in health facility TB treatment registers as cured, treatment completed, treatment failed, died, or lost to follow-up as per Uganda NTLP guidelines. The primary outcome was the proportion of patients with treatment success, defined as having a treatment outcome of cured or treatment completed recorded in the TB treatment register. Prespecified secondary outcomes included the proportion of patients completing the intensive phase of TB treatment (defined as completing 60 doses), the proportion of patients not lost to follow-up, and the proportion converted (proportion of bacteriologically positive patients who are smear-negative at 2 months). Conversion was removed as a secondary outcome after the trial began, as the data were not available; many trial sites had switched from smear microscopy to Xpert MTB/RIF as the primary method for TB diagnosis, and most trial sites did not routinely perform 2-month smear examination. In addition, within the intervention period, we assessed the proportion of patients who were enrolled on 99DOTS. Additional secondary outcomes related to the adoption and implementation of 99DOTS will be reported separately [8].

## Statistical methods

**Power and sample size.** With 18 clusters (health facilities), an anticipated harmonic mean of 15 eligible patients per month at each health facility, 6 steps (3 health facilities randomized to the intervention each month), and a type 1 error (alpha) of 0.05, the trial had 89% power to detect a 10% increase in the treatment success proportion, comparing the intervention relative to the control period [8]. The power calculation assumed a pre-implementation treatment success rate of 51% and an intraclass correlation coefficient of 0.001 (both determined using 2017 Uganda NTLP data from participating health facilities).

**Statistical analysis.** Primary and secondary outcomes were analyzed according to intention-to-treat (ITT) and per protocol (PP) principles. The ITT analysis included all eligible patients initiated on TB treatment during the control and intervention periods and thus aims to provide an unbiased estimate of the effect of the 99DOTS-based intervention in the study population overall. The PP analysis excluded (1) patients who initiated TB treatment while their health facility was in the control period but were later enrolled on 99DOTS at any point during treatment and (2) patients who initiated TB treatment while their health facility was in the intervention period but were not enrolled on 99DOTS during the first month of treatment. Although limited by the potential for selection bias, the PP analysis aims to estimate the effect of actually receiving the assigned intervention.

The main analyses were done using mixed effects logit models to estimate the adjusted odds ratios (aORs) and corresponding 95% confidence intervals, adjusting for clustering of observations at the health facility level as a random effect and time (trial month) and potential confounders (age, sex, HIV status, bacteriologically confirmed versus clinical TB diagnosis, new versus retreatment TB diagnosis, and level of health facility) as fixed effects.

For prespecified subgroup analyses stratified by sex and HIV status, we calculated aORs in the same manner. The intervention effect was compared among subgroups by assessing the significance of a subgroup–period interaction term when included in the non-stratified model. Adjusted proportions of each subgroup with the outcome and proportion differences associated with the intervention were output from the non-stratified model. A prespecified sensitivity analysis was conducted in which patients initiating TB treatment during the buffer period were assigned to the control period if they initiated treatment on or before the first day of 99DOTS training and to the intervention period if they initiated treatment after the first day of 99DOTS training. A post hoc sensitivity analysis included only patients with a phone number listed in the TB treatment register (as a proxy for phone access). For both sensitivity analyses, aORs were calculated in the same manner as for the primary analysis. Stata 15 was used for all analyses [12]. The trial is registered with the Pan African Clinical Trials Registry (PACTR201808609844917).

## Results

During the study period from 1 December 2018 to 31 July 2019, 2,790 adults initiated treatment for drug-susceptible pulmonary TB at the 18 health facilities, of whom 566 (20.3%) were ineligible for analysis because they transferred to other health facilities during treatment (S1 Fig). Within the control period, eligible patients were older (mean 39.1 versus 36.1 years, $p = 0.001$) than ineligible patients. Within the intervention period, eligible patients were older (mean 39.7 versus 35.5 years, $p < 0.001$), more likely to be HIV-positive (46.2% versus 37.6%, $p = 0.01$), and more likely to have bacteriologically confirmed TB (52.2% versus 41.6%, $p = 0.001$) than ineligible patients. There were no differences in the eligible versus ineligible populations by sex (37.0% versus 39.4% female, $p = 0.29$) or prior TB status (8.6% versus 9.2% retreatment, $p = 0.68$).

**Table 1. Participant baseline characteristics by study population and period.**

| Characteristic | Intention-to-treat population | | Per protocol population | | Buffer period (*n* = 311) |
|---|---|---|---|---|---|
| | Control period (*n* = 1,022) | Intervention period (*n* = 891) | Control period (*n* = 987) | Intervention period (*n* = 463) | |
| Age in years, mean (SD) | 39.1 (14.2) | 39.7 (14.6) | 39.2 (14.3) | 38.9 (14.2) | 39.4 (15.2) |
| Female, *n* (%) | 396 (38.8) | 330 (37.0) | 377 (38.2) | 167 (36.1) | 97 (31.2) |
| HIV-positive, *n* (%) | 484 (47.4) | 412 (46.2) | 469 (47.5) | 192 (41.5) | 123 (39.6) |
| On ART, *n* (%) | 480 (99.2) | 412 (100) | 465 (99.2) | 192 (100) | 123 (99.6) |
| New patient, *n* (%) | 936 (91.6) | 812 (91.1) | 902 (91.4) | 423 (91.4) | 283 (91.0) |
| Bacteriologically confirmed TB, *n* (%) | 498 (48.7) | 468 (52.5) | 480 (48.6) | 278 (60.0) | 164 (52.7) |
| Xpert-positive, *n* (%) | 474 (46.4) | 436 (48.9) | 456 (46.2) | 263 (56.8) | 157 (50.5) |

SD, standard deviation; TB, tuberculosis.

Of the 2,224 eligible patients, 311 who initiated TB treatment while their health facility was in the buffer period were excluded from the primary analysis (Fig 1). The ITT study population (*n* = 1,913) included the remaining 1,022 (53.4%) patients who started TB treatment while their health facility was allocated to the control period and 891 (46.6%) patients who started TB treatment while their health facility was allocated to the intervention period. The PP study population (*n* = 1,450) excluded 35 (3.4%) patients who were enrolled on 99DOTS while their health facility was allocated to the control period and 428 (48.0%) patients who were not enrolled on 99DOTS within 1 month of treatment initiation while their health facility was allocated to the intervention period. No patients asked for their data to be excluded, and no data were missing for the primary outcome analysis.

There were no differences in measured characteristics between patients who initiated treatment during the control and intervention periods, other than a higher proportion of patients with bacteriologically confirmed TB in the intervention period (52.5% versus 48.7%; Table 1). Overall, most (>91%) patients had newly diagnosed TB, mean (SD) age was 39.4 (14.4) years, 38.0% were women, and 46.8% were HIV-infected.

Within the intervention period, 463 of 891 (52.0%) patients were enrolled on 99DOTS within the first month of treatment. The remaining 428 were either never enrolled on 99DOTS (*n* = 387) or enrolled later on in treatment (*n* = 41). Patients enrolled on 99DOTS were similar to patients not enrolled on 99DOTS with respect to age (mean 38.9 versus 40.5 years, *p* = 0.11), sex (36.1% versus 38.1% female, *p* = 0.53), and history of prior TB (8.6% versus 9.1%, *p* = 0.80), but were less likely to have HIV infection (41.5% versus 51.4%, *p* = 0.003) and more likely to have bacteriologically confirmed TB (60.0% versus 44.4%, *p* < 0.001). The proportion enrolled on 99DOTS also varied across health facilities (median 56.3%, range 29.4%–72.2%, Levene's test *p* < 0.001). Reasons for non-enrollment on 99DOTS included lack of regular access to a phone (*n* = 277, 64.7%), patient preference (*n* = 33, 7.7%), death/loss to follow-up before enrollment (*n* = 28, 6.5%), being deemed too ill or elderly by health facility staff (*n* = 18, 4.2%), work- or school-related reasons (*n* = 5, 1.2%), and unknown (*n* = 67, 15.7%).

In the ITT analysis, 72.7% of patients in the intervention period and 70.9% of patients in the control period completed treatment successfully (Table 2). The monthly adjusted proportions of patients with treatment success were similar throughout the study period and overlapped between the control and intervention periods (Fig 2A). The 99DOTS-based intervention did not increase the odds of treatment success (aOR 1.04, 95% CI 0.68–1.58, *p* = 0.87). Similarly, the adjusted odds of persistence on treatment through the intensive phase

**Table 2. Effectiveness of the 99DOTS-based intervention.**

| Outcome and analysis | N | Outcome proportion, n/N (%) | | Effect estimate | | |
|---|---|---|---|---|---|---|
| | | Control period | Intervention period | Adjusted proportion difference†^ (95% CI) | Adjusted odds ratio† (95% CI) | p-Value‡ |
| **Treated successfully*** | | | | | | |
| ITT | 1,913 | 725/1,022 (70.9%) | 648/891 (72.7%) | 0.65 (−7.25, 8.55) | 1.04 (0.68, 1.58) | 0.87 |
| PP | 1,450 | 696/987 (70.5%) | 401/463 (86.6%) | 16.49 (7.66, 25.31) | 2.89 (1.57, 5.33) | 0.001 |
| **Completed intensive phase**** | | | | | | |
| ITT | 1,913 | 806/1,022 (78.9%) | 722/891 (81.0%) | 0.46 (−6.29, 7.22) | 1.03 (0.65, 1.63) | 0.89 |
| PP | 1,450 | 774/987 (78.4%) | 423/463 (91.4%) | 14.26 (7.03, 21.49) | 3.47 (1.71, 7.03) | 0.001 |
| **Not lost to follow-up**** | | | | | | |
| ITT | 1,913 | 892/1,022 (87.3%) | 780/891 (87.5%) | −1.06 (−6.85, 4.74) | 0.90 (0.52, 1.57) | 0.72 |
| PP | 1,450 | 857/987 (86.8%) | 448/463 (96.8%) | 10.83 (4.41, 17.25) | 4.92 (1.79, 13.49) | 0.002 |

ITT, intention-to-treat; PP, per protocol.

†Adjusted for time (trial month, discrete variable), sex, HIV status, disease class (bacteriologically confirmed versus clinically diagnosed), and TB type (new versus retreatment) as fixed effects and site as a random effect.

^ Proportion difference calculated as intervention minus control.

‡p-Value for adjusted intervention effect.

*Primary outcome.

**Secondary outcome.

(aOR 1.03, 95% CI 0.65–1.63) and not being lost to follow-up (aOR 0.90, 0.52–1.57) were similar in the intervention and control periods (Table 2).

In subgroup analyses, the effect of the 99DOTS-based intervention was strongest in men and HIV-positive patients. Within the ITT study population, the odds of treatment success were increased among men (aOR 1.24, 95% CI 0.73, 2.10) and HIV-infected patients (aOR 1.51, 95% CI 0.81, 2.85), but the aORs and the between-group differences for men and women ($p = 0.39$) and for HIV-infected and -uninfected patients ($p = 0.86$) were not statistically significant (Fig 3A).

In the PP analysis, 86.6% of patients in the intervention period and 70.5% of patients in the control period completed treatment successfully (Table 2). The monthly adjusted proportions of patients with treatment success were significantly higher in the intervention period compared to the control period (Fig 2B). The 99DOTS-based intervention increased the odds of

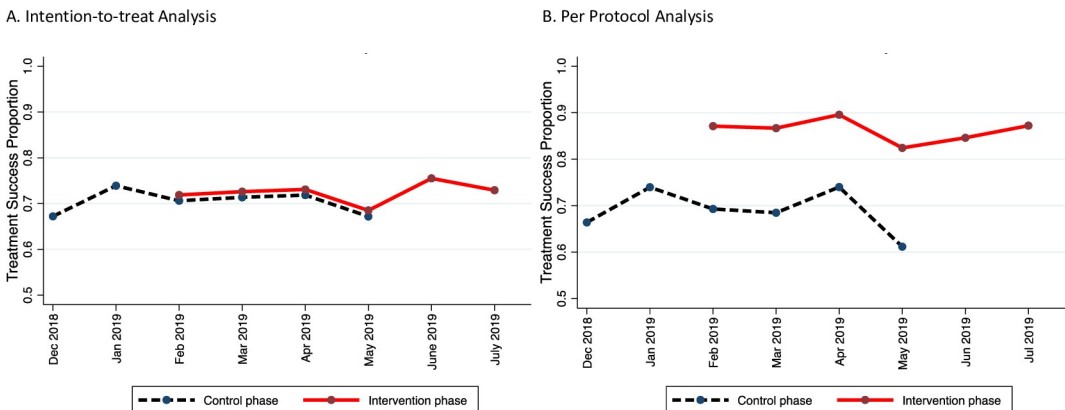

**Fig 2. Adjusted proportions of treatment success.** The data shown are adjusted proportions output by the primary analysis mixed effects logistic regression model for the (A) intention-to-treat analysis and (B) per protocol analysis.

## A. Intention-to-treat Analysis

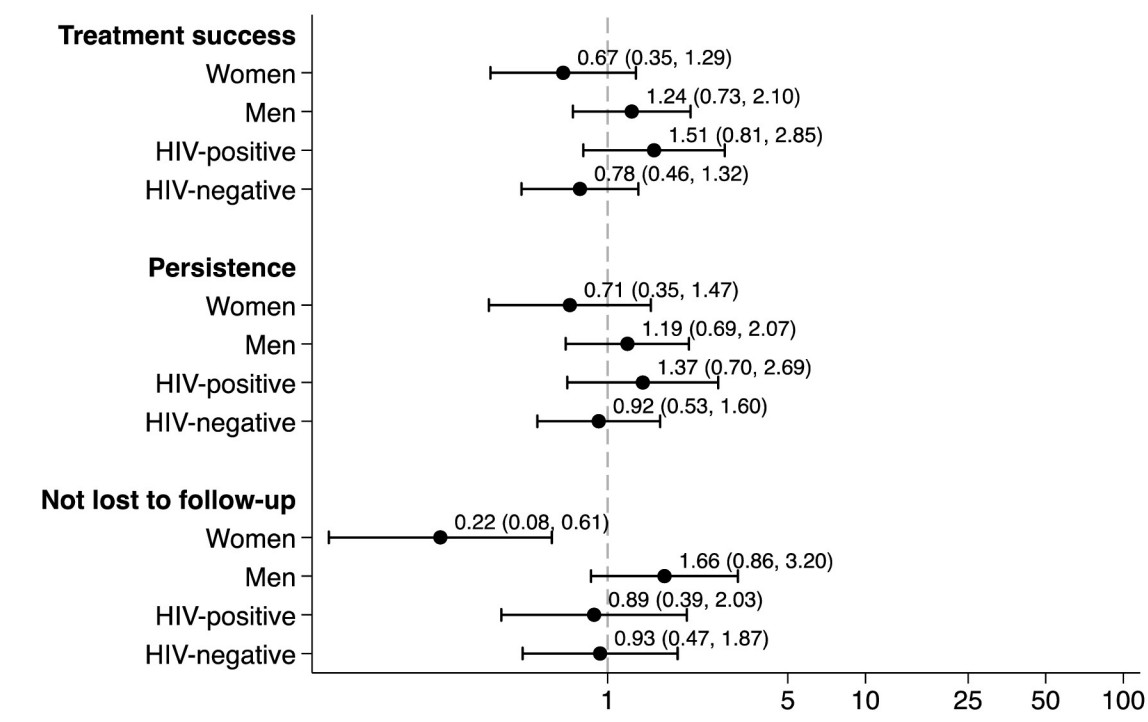

## B. Per Protocol Analysis

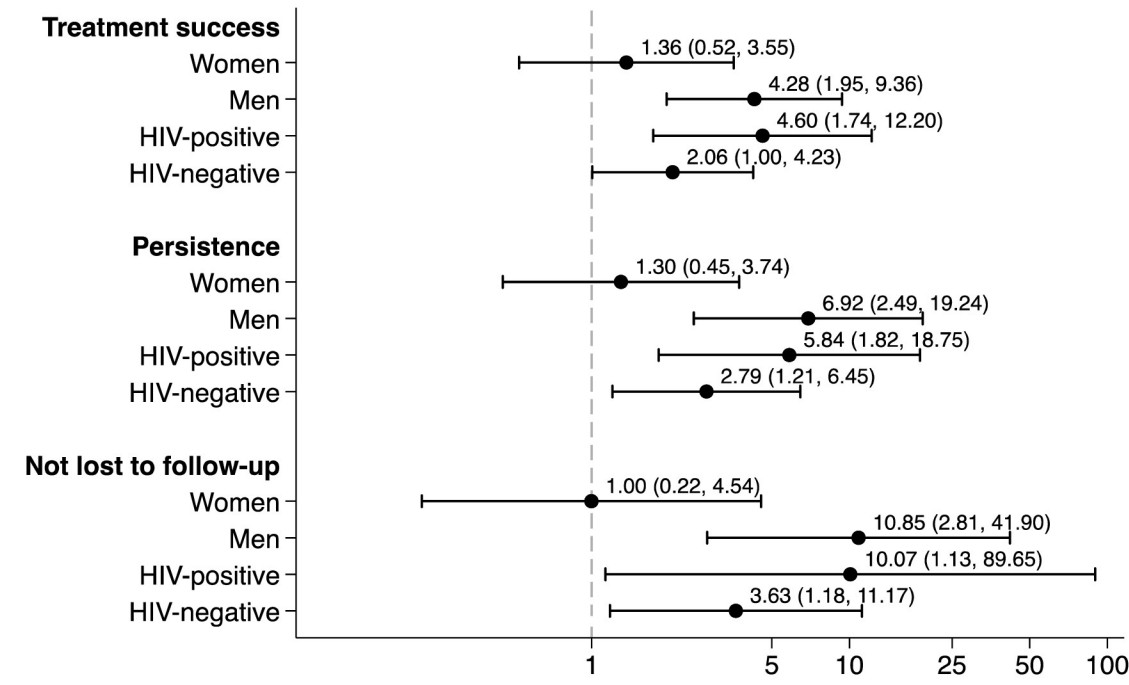

**Fig 3. Effectiveness of 99DOTS among subgroups.** The data shown are odds ratios (ORs) from mixed effects logistic regression models for the (A) intention-to-treat analysis and (B) per protocol analysis. All models are adjusted for time (month), disease class (bacteriologically

confirmed versus clinically diagnosed), and TB type (new versus retreatment) as fixed effects and site as a random effect. Sex-specific ORs are also adjusted for HIV status as a fixed effect, and HIV-status-specific ORs are also adjusted for sex as a fixed effect.

treatment success (aOR 2.89, 95% CI 1.57–5.33, $p < 0.001$), persistence on treatment through the intensive phase (aOR 3.47, 95% CI 1.71–7.03, $p = 0.001$), and not being lost to follow-up (aOR 4.92, 95% CI 1.79–13.49, $p = 0.002$) (Table 2). In subgroup analyses (Fig 3B), the odds of treatment success were significantly increased for men (aOR 4.28, 95% CI 1.95–9.36) and HIV-infected patients (aOR 4.60, 95% CI 1.74–12.20). As for the ITT population, the between-group differences for men and women ($p = 0.42$) and for HIV-infected and -uninfected patients ($p = 0.23$) were not statistically significant. The same patterns were observed in ITT and PP analyses for the subgroup effects of the 99DOTS-based intervention on persistence and loss to follow-up (Fig 3).

In a prespecified sensitivity analysis, the 311 patients from the buffer period were included in the control population if they initiated treatment on or before the first day of their health facility's 99DOTS training ($n = 118$) and in the intervention period if they initiated treatment after the first day of their health facility's 99DOTS training ($n = 193$). Similar to the main analyses, there was improvement in treatment success with 99DOTS in the PP analysis (aOR 2.62, 95% CI 1.66–4.12, $p < 0.001$) but not in the ITT analysis (aOR 1.16, 95% CI 0.83–1.61, $p = 0.39$) (S1 Table). In a post hoc sensitivity analysis including patients from the control period only if they had a phone number listed in the TB treatment register (as a proxy for access to a phone; $n = 473/1022$, 46.3%), the intervention effect was similar to that of the PP analysis (aOR 3.70, 95% CI 1.72–9.94, $p = 0.001$) (S2 Table).

## Discussion

In this stepped-wedge cluster-randomized trial of 1,913 patients with drug-susceptible TB in Uganda, there was overall no significant difference in outcomes between the control and intervention periods. Only 52% of patients in the intervention period were enrolled on 99DOTS-based treatment supervision; in the PP analysis, these patients were substantially more likely to complete treatment, including persisting on treatment through the intensive phase and avoiding loss to follow-up during treatment. Thus, while 99DOTS-based treatment supervision does not improve population-level treatment outcomes relative to DOT, it is likely a viable alternative to DOT for the substantial proportion of patients who have access to a phone and are interested in using this technology.

Of the 4 published randomized trials of DATs to support TB treatment [13–16], 2 were conducted in high-burden countries. In a large cluster-randomized trial in China [15], reminders from electronic medication monitors improved adherence (percentage of patient-months on treatment with <20% of doses missed) by 42%, and in an individual-level randomized controlled trial in Kenya [16], a custom digital health platform compatible with routine feature phones reduced unsuccessful treatment outcomes by 3-fold (13.1% versus 4.2%, $p < 0.001$). Previous observational studies of 99DOTS in India have reported variable acceptance by patients [17], suboptimal accuracy for measuring adherence [18], and worsening of treatment outcomes following its implementation [19]. Our trial confirms that 99DOTS—and likely other DATs—are unlikely on their own to substantially improve population-level TB treatment outcomes when implemented as part of routine care.

Due to the pragmatic nature of our trial, health workers and patients made all decisions about the use of 99DOTS during the intervention period, and only about half of all eligible patients were enrolled on 99DOTS. The substantially better treatment outcomes among these patients in the PP analysis should not be interpreted as improving outcomes in the overall

population, primarily due to the possibility of selection bias. Patients enrolled on 99DOTS likely represent a nonrandom sample of patients initiating treatment with higher interest in engaging in care and therefore higher likelihood of completing treatment even without 99DOTS. Although our post hoc analysis excluding patients without listed phone numbers showed similar findings to the PP analysis, it may not adequately control for this bias—the reason for non-enrollment on 99DOTS was unrelated to phone access for approximately one-third of patients in the intervention period. Despite these caveats, it is noteworthy that 99DOTS-based treatment supervision empowered half of all patients to take TB medicines at the time and place of their choosing, resulting in >85% treatment completion while mitigating the potential inconvenience, stigma, and costs commonly reported as barriers associated with DOT [20–22]. Further research is needed on whether overall treatment outcomes can be improved by increasing the uptake of low-cost DATs such as 99DOTS (e.g., by providing patients with phones or offering a choice of different DAT platforms) and strengthening the capacity of health workers to provide enhanced adherence support using real-time dosing information.

A major strength of our study is that it comprises a robust yet real-world evaluation of the impact of 99DOTS on TB treatment outcomes. The trial was conducted in TB treatment units similar to those found in other high-burden countries, and there were minimal exclusion criteria. The short training on the 99DOTS-based intervention [8] with subsequent patient management directed by routine health workers reflects how 99DOTS would eventually be scaled up. We therefore anticipate that these findings would be generalizable to public sector TB treatment units in other high-TB-burden countries.

Our trial also had several potential limitations. The highly pragmatic nature of the trial constrained our ability to control certain aspects of the design. For example, we were unable to collect adherence data for patients treated under routine care and did not verify the accuracy of patients calling to self-report medication dosing as a surrogate for actual dosing. However, the trial reflects treatment outcomes as they are routinely reported by national TB programs, and followed a prespecified protocol and analysis plan. The nature of the 99DOTS-based intervention precluded masking of clinicians, but study personnel (with the exception of the trial statistician and data manager) were blinded to aggregated treatment outcomes. As discussed above, the PP analyses should be interpreted with caution given the probability of selection bias in the PP population. Last, the 99DOTS-based intervention was not fully implemented as planned. In particular, aspects related to patient monitoring and support, including weekly interactive voice response check-ins and health worker task lists to facilitate patient follow-up, were not fully developed or implemented during the trial period, which could have reduced potential effectiveness.

In conclusion, our trial provides randomized evidence that 99DOTS-based treatment supervision does not improve population-level treatment outcomes. However, the high levels of treatment completion among those who used 99DOTS suggest that this technology may be a viable alternative to DOT for many patients. Given that 99DOTS likely reduces costs and burden to patients, these findings support a more patient-centered approach to TB treatment supervision that replaces universal DOT with offering 99DOTS-based treatment supervision as an alternative treatment supervision modality for those patients who are willing and able to use it.

## Supporting information

**S1 CONSORT Checklist.**
(PDF)

**S1 Data. Raw de-identified data used to conduct this analysis.** Each row corresponds to a patient. The dataset includes 13 columns: intervention group—randomization block (1–6); health facility—health facility where the patient initiated treatment (1–18); health center—type of facility (health center, hospital); trial month—month during which the patient initiated treatment (0–7); study period (control, buffer, intervention); sex; age; HIV status; disease class (bacteriologically confirmed pulmonary TB, clinically diagnosed pulmonary TB); retreatment—patient type (new, retreatment); treated successfully—primary outcome; persisting on treatment—secondary outcome; not lost to follow-up—secondary outcome.
(XLSX)

**S1 Fig. Health facilities included in the DOT to DAT trial.**
(TIF)

**S2 Fig. Stepped-wedge trial design and patient enrollment.** The target population includes all adults initiating treatment for drug-susceptible pulmonary TB. The eligible population excludes patients in the target population who were transferred out to another health facility during their treatment. Patients who initiated treatment during the buffer period were excluded from the study population.
(TIF)

**S3 Fig. Adapted 99DOTS platform.** The original 99DOTS envelope (top) had 2 sides. We redesigned the original envelope using human-centered design to reduce stigma, encourage appropriate dosing, and facilitate communication between patients and health workers. Prototype 1 (middle) was used from January to June 2019. Prototype 2 (bottom) was used from July 2019 through the end of the trial. In addition to the changes shown here, the ring tone heard when patients called toll-free numbers to self-report dosing was replaced with a rotating series of educational or motivational messages recorded by local health workers.
(TIF)

**S1 Statistical Analysis Plan.**
(DOCX)

**S1 Table. Prespecified sensitivity analysis of the effectiveness of the 99DOTS-based intervention when including the buffer period.** ITT, intention-to-treat; PP, per protocol. *Primary outcome. **Secondary outcome. †Adjusted for time (trial month, discrete variable), sex, HIV status, disease class (bacteriologically confirmed versus clinically diagnosed), and TB type (new versus retreatment) as fixed effects and site as a random effect. ^Proportion difference calculated as proportion in intervention period minus that in the control period. ¶$p$-Value for adjusted intervention effect.
(DOCX)

**S2 Table. Post hoc sensitivity analysis of the effectiveness of the 99DOTS-based intervention when excluding patients without a listed phone number.** *Primary outcome. **Secondary outcome. ‡Adjusted for time (trial month, discrete variable), sex, HIV status, disease class (bacteriologically confirmed versus clinically diagnosed), and TB type (new versus retreatment) as fixed effects and site as a random effect. ^Proportion difference calculated as intervention minus control. ¶$p$-Value for adjusted intervention effect.
(DOCX)

**S1 Trial Protocol.**
(DOC)

## Acknowledgments

We thank the administration, staff, and patients at participating health facilities. The DOT to DAT trial was implemented in collaboration with the Uganda NTLP and was made possible by the commitment of district TB focal persons and health facility staff.

## Author Contributions

**Conceptualization:** Adithya Cattamanchi, Noah Kiwanuka, Achilles Katamba.

**Data curation:** Rebecca Crowder, Alex Kityamuwesi, Maureen Lamunu, Catherine Namale, Lynn Kunihira Tinka, Agnes Sanyu Nakate, Joseph Ggita, Patricia Turimumahoro, Diana Babirye, Denis Oyuku, Christopher Berger, Austin Tucker.

**Formal analysis:** Adithya Cattamanchi, Rebecca Crowder, Noah Kiwanuka, Achilles Katamba.

**Funding acquisition:** Adithya Cattamanchi, Achilles Katamba.

**Investigation:** Adithya Cattamanchi, Noah Kiwanuka, Amanda Sammann, Stavia Turyahabwe, David Dowdy, Achilles Katamba.

**Methodology:** Adithya Cattamanchi, Rebecca Crowder, Alex Kityamuwesi, Noah Kiwanuka, Joseph Ggita, Christopher Berger, Austin Tucker, Devika Patel, Amanda Sammann, David Dowdy.

**Project administration:** Adithya Cattamanchi, Rebecca Crowder, Alex Kityamuwesi, Patricia Turimumahoro, Denis Oyuku, Christopher Berger, Devika Patel, Stavia Turyahabwe.

**Resources:** Stavia Turyahabwe.

**Supervision:** Alex Kityamuwesi, Austin Tucker.

**Visualization:** Rebecca Crowder, Noah Kiwanuka, David Dowdy, Achilles Katamba.

**Writing – original draft:** Adithya Cattamanchi, Rebecca Crowder, Alex Kityamuwesi, Noah Kiwanuka, Achilles Katamba.

**Writing – review & editing:** Adithya Cattamanchi, Rebecca Crowder, Alex Kityamuwesi, Noah Kiwanuka, Maureen Lamunu, Catherine Namale, Lynn Kunihira Tinka, Agnes Sanyu Nakate, Joseph Ggita, Patricia Turimumahoro, Diana Babirye, Denis Oyuku, Christopher Berger, Austin Tucker, Devika Patel, Amanda Sammann, Stavia Turyahabwe, David Dowdy, Achilles Katamba.

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
