## [Editor Report · Decision Letter 0]

9 Jan 2021

Dear Dr Cattamanchi, 

Thank you for submitting your manuscript entitled "Digital adherence technology for tuberculosis treatment supervision: a stepped-wedge cluster randomized trial" for consideration by PLOS Medicine.

Your manuscript has now been evaluated by the PLOS Medicine editorial staff as well as by an academic editor with relevant expertise and I am writing to let you know that we would like to send your submission out for external peer review.

Kind regards,

Thomas J McBride, PhD

Senior Editor

PLOS Medicine

---

## [Decision Letter · Decision Letter 1]

6 Feb 2021

Dear Dr. Cattamanchi,

Thank you very much for submitting your manuscript "Digital adherence technology for tuberculosis treatment supervision: a stepped-wedge cluster randomized trial" (PMEDICINE-D-20-06186R1) for consideration at PLOS Medicine. 

Your paper was evaluated by an academic editor with relevant expertise, and sent to independent reviewers, including a statistical reviewer. The reviews are appended at the bottom of this email and any accompanying reviewer attachments can be seen via the link below:

[LINK]

In light of these reviews, we will not be able to accept the manuscript for publication in the journal in its current form, but we would like to invite you to submit a revised version that addresses the reviewers' and editors' comments fully. You will appreciate that we cannot make a decision about publication until we have seen the revised manuscript and your response, and we expect to seek re-review by one or more of the reviewers. 

We hope to receive your revised manuscript by Mar 01 2021 11:59PM. Please email us (plosmedicine@plos.org) if you have any questions or concerns.

Please let me know if you have any questions, and we look forward to receiving your revised manuscript in due course. 

Sincerely,

Richard Turner, PhD

rturner@plos.org

Please add a new final sentence to the "Methods and findings" subsection of your abstract, which should begin "Study limitations include ..." or similar and quote 2-3 of the study's main limitations. 

Do you mean "waiver of informed consent" in the Methods section?

Please remove trademarks, e.g., in the Methods section. 

Throughout the text, please ensure that reference call-outs fall before punctuation, e.g., "... blister packs [5,6]." (noting the absence of spaces in the square brackets). 

Please remove the information on study funding from the "Acknowledgments" section at the end of the main text. Funding information should appear in the article metadata, via entries in the submission form. 

In the reference list, please ensure that journal names are abbreviated consistently, e.g., "PLoS Med.".

Are references 4 & 18 duplicates?

Please update reference 8. 

Please spell out the institutional author name for reference 13.

Please adapt the attached CONSORT checklist so that individual items are referred to by section (e.g., "Methods") and paragraph number rather than by line or page numbers, as the latter generally change in the event of publication. 

Please name the checklist "S1_CONSORT_Checklist" or similar, and refer to it by this label in your Methods section. Please also refer to the attached trial protocol document in the same way. 

Comments from the reviewers:

***Reviewer #1: 

This is an important and well-conducted stepped-wedge cluster randomized trial assessing the impact of implementing a digital adherence technology (DAT) on TB treatment outcomes in a low-income country. With ongoing critique of DOT models in TB care, interest in DATs—as a replacement for DOT—has been growing; however, there is a paucity of evidence regarding the impact of these technologies on TB treatment outcomes and longer-term outcomes, particularly in low- and middle-income country settings. As such, this study is a potentially important contribution.

While the study is well-conducted, I have considerable concerns about the interpretation and framing of the study findings, particularly with regard to the per protocol analysis, which seems to drive the entire Discussion section of the paper. While the authors appropriately interpret the findings of the intent to treat analysis—that 99DOTS-based treatment supervision did not improve overall treatment outcomes in the patient population—the very substantial limitations of the per protocol analysis raise questions about their interpretation that "99DOTS improved treatment outcomes among patients who received the intervention." I worry that, given the considerable limitations of the per protocol analysis, this finding could be very misleading, especially given the lack of transparent discussion of these limitations—which I believe are significant enough that they should be emphasized in the abstract, author summary, and discussion sections, so as to signal to the reader that the study findings have not ultimately answered the question of whether this intervention improves outcomes. I outline the reasons for my concerns further below.

Major feedback:

1. Interpretation of the per protocol analysis findings: Per protocol analyses have the potential to lead to erroneous findings of intervention effect because—particularly when the analysis excludes a large number of patients that were in the intent to treat analysis—it breaks the random assignment that a randomized trial is supposed to achieve. As such, the correct interpretation of these findings is not necessarily that outcomes improved only among those who received the intervention, but rather that—given the exclusion of a large number of the original patients included in the trial—it remains unclear whether the intervention caused the improved outcomes, or whether the patients left behind in the per protocol analysis are those who were likely to have better outcomes anyways even without the intervention (i.e., introduction of selection bias). In the Discussion section, the authors allude very briefly to this limitation of their study before minimizing it when they state: "It is possible that the patients who were enrolled on 99OTS would have been more likely to complete the treatment even under routine care."

I worry that this is a much bigger problem than they suggest, for the following reasons:

(a) Per protocol findings may be more likely to reveal a real effect if there is at least some concordance in the directionality of the finding in the ITT analysis (even if the ITT findings were not significant). However, the ITT findings here don't have the same direction (the odds ratio nicely straddles 1) as the PP analysis findings.

(b) PP findings may be more likely to reveal real effect if it excludes a small proportion of patients; however, this PP analysis excluded nearly half (48%) of study participant in the intervention arm, while only about 3% were excluded from the control group, which could introduce a high level of bias into the findings, as the PP intervention group essentially may represent a non-random sample of the intervention group.

(c) Further the discretion exercised by both healthcare workers and patients in deciding which patients should be enrolled in 99DOTS (from the healthcare workers' side) and whether to engage with 99DOTS (from the patients' side) suggest a possibility that patients who enrolled in 99DOTS may have had higher baseline interest in engaging in care and may have been likely to have better outcomes even without the intervention. As such, when the PP analysis restricts to this subset of patients in the intervention group, it is hard to know whether the effect seen is a result of the intervention or of selection bias in the PP analysis. This is of course not the fault of the authors or the study design - but it does speak to the importance of placing precedence on the findings from the ITT analysis (which showed no effect), and of interpreting the PP analysis cautiously, rather than suggesting it shows a clear treatment effect. It's not clear if the PP analysis has much validity given the loss of half of the intervention sample.

(d) In addition, 6.5% of those who didn't enroll in 99DOTS died or were lost to follow-up before they could enroll and another 4.2% were deemed too ill or elderly to enroll - presumably they weren't included in the PP intervention group. The 6.5% all clearly had bad outcomes while the 4.2% were highly likely to have bad outcomes - and, from what I can tell, early deaths or severely ill individuals were not excluded from the PP control arm. This is a very clear source of bias in favor of seeing a positive effect from the 99DOTS/intervention group. In fact, just the exclusion of this ~10% of patients from the intervention period group could account for much of the ~10% difference in outcomes seen between the control and intervention periods in the PP analysis. Again, highlights the concerns about the PP analysis and the importance of highlighting these limitations and prioritizing the finding from the ITT analysis.

(e) Finally, the authors do a post-hoc analysis in which they compare the individuals in the intervention period/group in the PP analysis to those in the control/period group who had phone numbers listed on their treatment cards, in a reasonable attempt to control for the bias in the PP intervention group. However, their description of patients who engaged with 99DOTS suggests that it may not match well with this subset of patients in the control period. For example, 65% of patients who didn't enroll in 99DOTS in the intervention period didn't have regular access to a phone - which suggests one-third of patients actually did have regular access to a phone but still chose not to enroll in 99DOTS, again raising concerns about those who enrolled with 99DOTS being more engaged with their TB care even without the intervention. As such the intervention period/group in the PP analysis excludes many patients who actually had regular access phones. Also, in the modified PP control group, just having a phone number listed on the treatment card does not mean that these patients will have regular access to their phones - which is another reason to why this modified PP control group probably does not actually address the problem of bias in the PP analysis.

2. Importance of IIT findings versus PP: Everything described in major critique #1 above does not undermine the trial study design per se - it just highlights that the primary valid finding is the finding from the ITT analysis, and the one from the PP is interesting but at such high risk of bias that using it as the focus of the paper could be misleading in suggesting 99DOTS has a treatment effect when it may or may not. The framing is critical. If the discussion is framed around the ITT findings, the reader walks away viewing this as an intervention with no clear benefit to the overall population of TB patients, but that may still be worth investigating further. Such a framing, which is more appropriate in my mind, supports further research rather than a "scaling up" of 99DOTS as advocated by the authors. The current discussion framed around the PP findings has readers walking away thinking that this intervention has clear benefit to those who use it, when this is far from clear given the substantial limitations and bias in the PP analysis - and therefore the current discussion is misleading. The ITT findings are clearly the scientifically sound findings.

In addition to reframing the discussion, a robust discussion of the limitations and biases in the PP analysis should be included in the Discussion section.

Minor feedback:

1. On page 5, the authors state that "The most common DAT platform…within an electronic pillbox." Electronic pillboxes certainly are being used at scale in China, but there are many other DATs in wide use - including video DOT in high-income settings. In fact, the most commonly used DAT is possibly 99DOTS, given its very large-scale rollout to hundreds of thousands of patients in India, Myanmar, and elsewhere. As such, revise this to say "A common DAT platform…"

2. On page 18, "To our knowledge, these data represent the first to show the impact of a low-cost DAT on TB treatment outcomes." - This isn't a correct statement, as the authors note in the very same paragraph. The Yoeli et al. trial of a two-way SMS platform in Kenya that the authors discussed showed an decline in unfavorable treatment outcomes (and therefore, conversely, an improvement in favorable outcomes). Also, if the authors take major critique #1 above seriously, then this also would not be a true statement b/c it is not clear that this study shows a real impact on treatment outcomes, if they prioritize the ITT findings.

3. There is some observational literature on TB treatment outcomes with 99DOTS in India - see Thekkur et al. Global Health Action 2019;12(1):1568826. The study finds poorer treatment outcomes at sites that rolled out 99DOTS, even after some adjustment for differences in patient characteristics. This is of course an observational study with its own risk of bias - however, there is similar risk of bias in the current PP analysis given the large proportion of patients excluded in the intervention period. It would be helpful for the authors to discuss their study findings in the context of this prior research on 99DOTS. There is also research on 99DOTS accuracy for measuring adherence and patient engagement with the technology that might be helpful to contextualize their findings.

*** Reviewer #2: 

Improved tuberculosis (TB) treatment outcomes were observed in a controlled trial of over 2200 participants in Uganda in 2018-2019 when patients contacted their healthcare giver on a freephone service after taking medications, as against in person direct observation of treatment. The article includes novel findings from the study of 99DOTS, a low-tech digital adherence technology that has been implemented at scale in India and elsewhere ahead of completion of trials. There is thus a lot of interest in the finding of this RCT (esp. during lockdowns and also for the delivery of other treatments, like ART). Moreover the adaptations to the basic technology that were introduced by the investigators of the Uganda trial (educational messages to callers and more neutral packaging) appear to be helpful and could be integrated in the basic package of 99DOTS. In my opinion this work can be published subject to some minor revisions to improve understanding and the authors should be allowed a reasonable excess over the word limit to explain certain key issues of their interesting research and its implications adequately:

page 2, line 1: treatment completion for new & relapse cases is on target globally (at around 85%). Maybe you can state that in many countries (including Uganda) adherence and treatment completion remain problematic and that loss to follow up is one of the eminent barriers to improving cure in TB patients 

page 6, last sentence: it would be critical to add some details from the article referenced about the extent of local DOT implementation around the time that the trial was conducted. The reader needs to understand how much the SOC in Uganda matches what should be happening on paper

page 11, bottom lines: "However, treatment success even among patients using 99DOTS was below the 90% target specified in the END TB Strategy." isn't this partly explained by your inclusion of centres that had a treatment success <80% in previous years (the mean was only 51% treatment success as per Methods in p8)? the national reported rates of 74% in new & relapse cases in 2018 may mean that quite a few centres could reach the target values with the support of aids like 99DOTS (https://worldhealthorg.shinyapps.io/tb_profiles/?_inputs_&entity_type=%22country%22&lan=%22EN%22&iso2=%22UG%22) ? can you comment some more in the Discussion about this?

page 12, para 2: here or in the last para on this page can you add one or two sentences to explain why the stepped-wedge design is expected to provide a better approach to the study of DAT when compared to other more standard methods? if you believe there are downsides please add a note in the limitations para further down. Was there risk of "contamination" between participants in the intervention and control groups when they returned to the clinics to collect medication during follow up in centres randomised to the intervention in the first months?

page 12, midway: "Our trial extends upon these prior findings and lends further support to the increasing calls to abandon DOT and make the TB treatment experience more patient-centered" can you temper this commens with the relative disadvantages of 99DOTS as an approach when compared with direct physical interaction with a healthcare worker? in your opinion, and based on the findings, which patient risk factors and at which stage in their treatment would you think this DAT could be particularly helpful (e.g. would you posit that the improved effect in HIV-positive and in men has application beyond your study population)? do you anticipate an intervention model whereby all TB patients get 100% 99DOTS as the primary standard of care?

page 13, conclusion para: do you think that the adaptations explained in supplementary figure 2 contributed to the effects observed and do you advise similar customization in the future implementation of the technology?

*** Reviewer #3: 

This study reported a stepped wedge clustered randomized trial aiming to evaluate the effectiveness of 99DOTS in improving treatment outcome among drug susceptible tuberculosis (TB) patients among 18 health facilities in rural Uganda. This is an important study to fill the gap of the lack of evidence of applying the new digital technologies in improving TB treatment adherence in low and middle income countries. The study reported a non-significant result on the primary outcome, i.e., treatment success rate, between patients under 99DOTS intervention and those in routine care based on the intention to treat (ITT) analysis, but significant improvements in the intervention group compared with control regarding primary, secondary outcomes and in sub-group analyses. The manuscript was well written and clearly presented, but needs substantial work to address the following methodological concerns regarding patient inclusion, analyses and discussion.

1. Per protocol analysis was not mentioned in the protocol. This must be an ad-hoc analysis and have to be stated clearly the rationale of the PP analysis, and how the criteria of per protocol analysis was established. 

2. There are other alterations among the secondary outcomes, comparing what was reported on page 17 of the protocol. In the manuscript, the nominator of one of the secondary outcomes, the proportion of persistence, has been changed from patients completed at least 60 doses of treatment to patients who completed the intensive phases. There were no reports on the proportion converted as in the protocol.

3. What are the adherence rates of patients on 99DOTS program? Did non-adherence happen not only in the beginning but also in other periods during the treatment, e.g., when patients become better by the end of the treatment? Within the PP analysis, why the study excluded patients who did not enrol on 99DOTS within the first month, but did include patients who stopped using or dropped out 99DOTS in subsequent months?

4. On Page 7 of the manuscript, the authors stated that patients who started TB treatment during the buffer period, i.e., the first month of switching from control to intervention, were excluded from primary analysis. However, in S1 Figure, the flow chart, the authors did not state the number of patients excluded for this reason. This should appear in the same line as the 566 transferred-out patients.

Table S4 indicated that patients who initiated their treatment during the buffer period were not excluded from analysis (probably included for the control period), because the total number of patients, if including those initiating treatment during the buffer period, are the same (n=2224) as before. This seems to contradict the methods section.

If these patients were included in the control period, this will dilute the difference between the two groups.

5. There have been a large number of patients who did not enrol in 99DOTS program in the first month (47%, 428/ 891). What were reasons the authors may identify from the process evaluation? These patients apparently had much worse outcomes compared with the rest of their peers in the intervention group/period. This is important message that needs to be discussed.

6. In Table S5, there is a large number of patients who did not register a telephone number in the records. Were these patients provided with a cell phone? On page 19, the authors reported that 99DOTS was not used for a substantial proportion of the patients. What is the rate of patients using the 99DOTS? There are a number of related outcomes in the category of Implementation outcomes. This needs to be either reported or quoted from your other publications. This a crucial to understand the effectiveness of 99DOTS. It seems that the author indicated that the acceptability/ usability of 99DOTS were low. So the program is only effective among patients who use it.

I would suggest the authors would focus on the possible reasons regarding the difference between the ITT and PP analyses in the discussion section. 

*** Reviewer #4: 

This is a well-conducted stepped-wedge cluster RCT on digital adherence technology for tuberculosis treatment supervision. The study design, dataset, statististical methods and analyses, and presentation (tables and figures) and interpretation of the results are mostly adequate and of a good standard. However, there are still some issues needing attention.

1) Sample size calculation. With the information on Page 10, one is unable to reproduce the sample size and power. More details of the sample size calculation (parameters) are needed. How many clusters in the calculation? is it 18 or 6? How many steps? what's the cluster size? what's justificaiton of the assumption of 10% increase in the treatment success proportion? Where is the ICC of 0.001 from and any justification? Stepped-wedge design is complex so that sample size calculation needs to be very clear and detailed.

2) The ITT vs PP analysis is the key of the paper. However, the definiton of the PP analysis "The PP analysis excluded patients who initiated treatment during the control period but were nonetheless enrolled on 99DOTS prior to treatment completion" is a bit difficult to understand. Can authors please explain what exactly this means? 

*** Reviewer #5: 

Thank you for the opportunity to review this interesting submission. I have a number of comments and suggestions, but I do believe that this paper presents information important to the discipline (improving TB treatment/health outcomes through the use of innovative, patient-centered technologies) and is meritorious of publication. The paper correctly points out that, while there are many studies and publications addressing the use of digital adherence technologies to improve medication adherence, there are relatively fewer studies and publications (i) involving TB, (ii) conducted in highly pragmatic settings with a view toward generalizability with respect to other high burden regions, and, most importantly, (iii) focused on treatment/health outcomes as a primary endpoint. Especially given the other aspects of this study that have been published (user-centered design aspects) and are indicated as to be published (cost-effectiveness aspects), this study/paper represents the centerpiece of what is probably the most comprehensive evaluation to date of the use of digital adherence technologies as an enabler/enhancer of TB care. Again, I think it merits publication.

My specific comments and questions to the authors are set forth below:

1. My most important question/comment related to the primary end point and the distinction drawn between ITT and PP populations. I understand the distinction and the figures did an OK (not great job) of showing the differences in these patient populations. However, given the extremely important differences in outcomes (no significant effect in ITT and quite significant effect in PP), I think that the authors should be clearer about which is the more relevant group/more significant finding and why. It would seem that one could argue that the PP population/finding is more important in that more patients in this group seem to have received the actual intervention in the intervention arm. However, the PP group also excludes a number of "typical" patients so maybe it is an unrepresentative population. I would like to see the ITT/PP distinction and significance made clearer if possible. The authors stater on page 28 that "in conclusion, our trial provides evidence from a randomized trial that 99DOTS-based treatment supervision is effective for adults with pulmonary TB who are enrolled on the platform." Why isn't that the true bottom line here, rather than looking to an ITT population where the results seem more affected by "reach" than "efficacy" of the intervention? 

2. Related to point #1 above, I found the abstract to be underwhelming (compared to the well-stated conclusions and insights on pages 19-20). I wonder if some improvements might be possible/in order.

3. On page 6 of the paper, there is a reference to $4-6/patient at scale. Have the authors validated that pricing in Uganda or is this based on 99DOTS or other information? I am not sure the parenthetical adds value (maybe low-cost alone is sufficient) and may be provocative if not substantiated as real in the Uganda context.

4. I very much like the secondary end points related to persistence through intensive phase and also loss to follow up. Given the potential impact of these aspects on both transmission and recurrence, I applaud the study design and inclusion of these additional end points.

5. I wonder if it is worth noting in the discussion and conclusions that Uganda seems to have primarily community as opposed to facility-based DOT. That is noted earlier in the paper, but it might be worth noting that the conclusions supporting use of 99DOTS vis a vis DOT are in that (community-based rather than facility-based) context.

6. I thought the description of the 99DOTS intervention was clear. I wonder if it might be worth adjusting language slightly to highlight that the "hidden number" is a feature designed to increase confidence that the patient has "engaged" with his/her medication and therefore the dosing signal is more accurate. Not a major point.

7. I did have to get out a calculator to determine how the populations in the ITT and PP groups were calculated. I wonder if a figure showing how these groups were derived and the various exclusions applied in getting to the PP group might be helpful.

8. Again, just to reinforce the point -- the paragraph on page 15 is a very strong statement of the effectiveness of 99DOTS (in treatment outcomes, in persistence through IP phase, and in LTFU -- 3X more likely to complete treatment) for the PP group. If that is really the more relevant group, these findings are really important. But is it -- or is the ITT group the more relevant. Unclear.

9. The Kenya study referenced on page 18 had, I believe, a health outcomes impact/end point. Are the authors comfortable with the characterization on page 18?

10. On page 19, the authors state: " Even though overall treatment outcomes did not improve . . . " and then goes on to talk about other, ancillary benefits of 99DOTS (empowerment, etc.) Is that statement correct and/or appropriate? Elsewhere in the paper the authors make it clear that 99DOTS did in fact improve outcomes for patients enrolled thereon. Should this statement on 19 be revised? IF the ITT result was more about reach than effectiveness (do the data give any insights on this?), then that is not clear -- especially in statements like the one referenced. 

I hope these comments are helpful, and thanks to the authors for their efforts on behalf of TB patients globally.

***

[LINK]

---

## [Decision Letter · Decision Letter 2]

6 Apr 2021

Dear Dr. Cattamanchi,

Thank you very much for re-submitting your manuscript "Digital adherence technology for tuberculosis treatment supervision: a stepped-wedge cluster randomized trial" (PMEDICINE-D-20-06186R2) for consideration at PLOS Medicine.

I have discussed the paper with editorial colleagues and our academic editor, and it was also seen again by three reviewers. I am pleased to tell you that, provided the remaining editorial and production issues are fully dealt with, we expect to be able to accept the paper for publication in the journal.

[LINK]

Please let me know if you have any questions, and we look forward to receiving the revised manuscript shortly   

Kind regards,

Richard Turner, PhD

rturner@plos.org

Requests from Editors:

Please quote the study setting (Uganda) in your title. 

Please quote aggregate demographic details for participant groups in the abstract.

We ask you to de-emphasize the per-protocol findings in your abstract and main text. In the abstract, we suggest quoting the ITT findings, followed immediately by a summary of the subgroup findings by sex and HIV status (noting that these are non-significant), with the per-protocol findings summarized briefly thereafter. 

Please avoid claims such as "the first" (e.g., in your author summary) and where needed add "to our knowledge" or similar. 

Please trim the author summary so that each subsection contains 3-4 short points. 

Similarly, in the main text please present the subgroup analyses before the per-protocol findings. 

Throughout the text, please use the style "the 4 randomized trials" (though numbers should be spelt out at the start of sentences). 

Comments from Reviewers:

*** Reviewer #1: 

I have reviewed the changes made to this manuscript, and the authors have done a very nice job of addressing my main concern, which was the interpretation and claims made regarding the findings of the PP analysis.

The authors have shifted the main take-away of their study to suggest that 99DOTS may be a viable alternative to DOT for some patients, while also affirming that the intervention did not improve outcomes in the overall patient population. While they did not use a non-inferiority trial design, I think that this is a reasonable conclusion given comparable outcomes during the control and intervention periods, as well as the generally good outcomes among the non-random subset of patients who used 99DOTS. Of course, this conclusion may just reflect the many shortcomings of DOT as a model of care, rather than any actual benefits of 99DOTS.

My only final piece of feedback is that, given that the authors suggest that 99DOTS may be a viable alternative to DOT for some patients in this setting, they should provide a better description of what the "control" condition of DOT was that they were actually comparing against. "Community DOT" can mean many different things, and the authors note that there is considerable variability in implementation of community DOT. With that said, it would still be critical for readers to know: 

(1) In general (acknowledging that there may be variability) who most commonly serve as treatment supporters? Are they patients' family members—such that it is relatively convenient/easy for patients to be observed? Are they other people living in the community? Are they local healthcare workers?

(2) Are treatment supporters paid or compensated by the government?

(3) Are there any other costs of DOT in this context that would not also be there under 99DOTS-based monitoring? (Authors do not need to provide the exact costs but just very briefly—in a few words—note other aspects of care where costs may be incurred that might be different with DOT as compared to 99DOTS).

Two or three sentences in the Methods better describing what community DOT means in this context—even while recognizing that the model may be variable—is important for readers to be able to assess the authors' final claim that 99DOTS may help some patients avoid the "inconvenience and additional costs of directly observed therapy." 

The authors don't provide enough basic information for the reader to assess whether this could potentially be true. For example, if "treatment supporters" are simply unpaid family members, this would essentially be free for the government and therefore cheaper than 99DOTS, which involves extensive printing of paper envelopes, personnel time to pack medication blister packs in these envelopes and review the dashboard, tech support costs, lost time for patients to call phone numbers on a daily basis, and other costs. On the other hand, if treatment supporters are usually paid non-family members then it would be clear that 99DOTS may potentially be cheaper to the government and more convenient for some patients, as the authors claim.

This clarity would be very helpful to readers. A major problem with the TB digital adherence technologies (DATs) literature is that the "control" condition can be so variable across contexts and is often poorly defined in papers that are trying to assess whether these improve outcomes.

Otherwise, the authors have done an excellent job of addressing my concerns and those of other reviewers and I have no other concerns. Thank you for involving me in the peer review of this manuscript.

*** Reviewer #3: 

I felt the authors have adequately addressed questions, and made substantial revision of the manuscript. 

*** Reviewer #4: 

Thanks authors for their effort to improve the manuscript. I am satisfied with the response and revision. No further issues needing attention.

***

[LINK]

---

## [Editor Report · Decision Letter 3]

14 Apr 2021

Dear Dr Cattamanchi, 

On behalf of my colleagues and the Academic Editor, Dr Suthar, I am pleased to inform you that we have agreed to publish your manuscript "Digital adherence technology for tuberculosis treatment supervision: a stepped-wedge cluster randomized trial in Uganda" (PMEDICINE-D-20-06186R3) in PLOS Medicine.

In the author submission form, the data statement and competing interest statements, for example, appear to be duplicated - this will need to be rectified prior to final acceptance. One iteration of the competing interest statement quotes a "DS", presumably an error as no author has these initials. 

PRESS

Sincerely, 

Richard Turner, PhD 

rturner@plos.org